# Human Cytomegalovirus (HCMV) Genetic Diversity, Drug Resistance Testing and Prevalence of the Resistance Mutations: A Literature Review

**DOI:** 10.3390/tropicalmed9020049

**Published:** 2024-02-15

**Authors:** Ivana Grgic, Lana Gorenec

**Affiliations:** Department of Molecular and Immunological Diagnostic, University Hospital for Infectious Diseases “Dr. Fran Mihaljevic”, 10000 Zagreb, Croatia

**Keywords:** human cytomegalovirus, drug resistance, drug-resistance testing

## Abstract

Human cytomegalovirus (HCMV) is a pathogen with high prevalence in the general population that is responsible for high morbidity and mortality in immunocompromised individuals and newborns, while remaining mainly asymptomatic in healthy individuals. The HCMV genome is 236,000 nucleotides long and encodes approximately 200 genes in more than 170 open reading frames, with the highest rate of genetic polymorphisms occurring in the envelope glycoproteins. HCMV infection is treated with antiviral drugs such as ganciclovir, valganciclovir, cidofovir, foscarnet, letermovir and maribavir targeting viral enzymes, DNA polymerase, kinase and the terminase complex. One of the obstacles to successful therapy is the emergence of drug resistance, which can be tested phenotypically or by genotyping using Sanger sequencing, which is a widely available but less sensitive method, or next-generation sequencing performed in samples with a lower viral load to detect minority variants, those representing approximately 1% of the population. The prevalence of drug resistance depends on the population tested, as well as the drug, and ranges from no mutations detected to up to almost 50%. A high prevalence of resistance emphasizes the importance of testing the patient whenever resistance is suspected, which requires the development of more sensitive and rapid tests while also highlighting the need for alternative therapeutic targets, strategies and the development of an effective vaccine.

## 1. Introduction

Human cytomegalovirus (HCMV) belongs to the Herpesviridiae family and the Betaherpesvirinae subfamily. It causes severe clinical disease with multi-organ involvement and frequent fatal consequences in immunocompromised individuals. Iatrogenic, like transplant patients, or acquired, such as HIV infected subjects. HCMV infection is also the most common vertically transmitted infection and a major cause of birth defects in newborns. Primary infection, reinfection or reactivation in immunocompetent individuals are usually asymptomatic [1,2]. Seroprevalence in the general population ranges from 45 to 100% and tends to be highest in South America, Africa and Asia and lowest in Western Europe and the United States. Hemodialysis patients are at the highest risk of being infected, but seropositivity is also age-dependent with prevalence ranging from 53.0% in children less than 10 years old to 93.8% in persons above the age of 60 [3,4]. Consistent with the diverse symptoms of the disease, HCMV can infect almost all cell types, with a strong tropism for fibroblasts, epithelial, endothelial, smooth muscle and placental cells. It causes lifelong latent infection after primary infection by establishing latency in long-lived cell populations, resulting in lytic viral replication and potential periodic reactivation from latency or reinfection with a new strain [5,6].

## 2. The HCMV Genome and Genetic Diversity

The HCMV genome is a 236 kb long linear double-stranded DNA (dsDNA), which is longer than all other human herpesviruses and one of the longest genomes of all human viruses in general. The genome contains more than 170 open reading frames and encodes approximately 200 genes, including nine gene families, a large number of glycoprotein genes, and homologues of the human HLA class I and G protein-coupled receptor genes [7,8,9,10]. The genome has the highest level of genetic variability of all the known human herpesviruses. The virus is known to readily undergo recombination, and coinfection is frequently observed, especially in individuals with weakened immune systems [11,12,13]. HCMV clinical isolates display genetic polymorphisms in multiple genes, mainly envelope glycoproteins such as UL55 (gB), UL73 (gN) and UL75 (gH) [13]. Glycoprotein B (gB), encoded by the UL55 gene and classified into 5 genotypes (gB1, gB2, gB3, gB4 and gB5), is an abundant and the most highly conserved glycoprotein of CMV. The glycoprotein H (gH), divided into two major genotypes (gH1 and gH2), is an 86 -kDa protein and encoded by the UL75 gene. The highly polymorphic gene UL73 encodes the viral glycoprotein N (gN), which is divided into seven genotypes: gN1, gN2, gN3a, gN3b, gN4a, gN4b and gN4c [14,15,16,17]. Past research has attempted to tie certain polymorphisms to the higher viral fitness of the strain, and, as a consequence, to different clinical manifestations of HCMV disease, and the ability to establish persistent or latent infections, but every attempt to correlate individual alleles with transmission and pathogenesis have so far been unclear or contradictory [18,19,20,21,22].

## 3. HCMV Antiviral Therapy

Antiviral agents specifically targeting HCMV-like ganciclovir (GCV), valganciclovir (VGV), foscarnet (FOS) and cidofovir (CDV) interfere with the synthesis of viral DNA by binding to the active site of the viral DNA polymerase (UL54) [23].

### 3.1. GCV and VGV

The frontline drugs for the treatment of HCMV infection and prophylaxis, GCV and its oral prodrug VGV, exhibit only modest antiviral activity that is often insufficient to completely suppress viral replication and drives the selection of drug-resistant variants to continue to replicate and contribute to disease. To obtain to its active form, ganciclovir-5′-triphosphate, a nucleoside analog that targets DNA polymerase, GCV undergoes phosphorylation by both cell and viral kinase (UL97). GCV is most frequently administered as an intravenous formulation due to its low oral bioavailability. VGV is a GCV ester that is well absorbed after oral administration and rapidly metabolizes to ganciclovir. Both drugs are routinely used for treating HCMV infection after solid organ transplantation, but due to possibility of myelosuppression, it is advised to avoid administration after hematopoietic cell transplantation [24,25]. 

### 3.2. FOS

FOS, a pyrophosphate analogue, also has increased affinity for UL54 and blocks DNA replication, but does not require phosphorylation; therefore, resistant mutations can only be formed in viral DNA polymerase genes [26]. It is not the drug of choice in first-line preemptive therapy for HCMV due to its considerable nephrotoxicity, and it is administered only if a patient is cytopenic, or resistance to first-line therapy agents has been proven. 

### 3.3. CDV

CDV is a cytidine monophosphate analogue and a competitive inhibitor of the viral DNA polymerase that undergoes phosphorylation using cell kinase so that the resistance mutations can be detected in the UL54 gene. It is also reserved for second-line treatments due to its considerable toxicity [27,28]. 

### 3.4. Maribavir

Maribavir (MBV) is an oral benzimidazole nucleoside that effects HCMV DNA synthesis, viral gene expression, encapsidation and viral capsid egress through the inhibition of the UL97 kinase. The United States Food and Drug Administration approved it in November 2021 for the treatment of adult and pediatric refractory/resistant post-transplant HCMV infection [29,30,31]. Resistance mutations to MBV emerge in the UL97 gene, which may result in cross-resistance to GCV and exclude the option of combination therapy [32,33,34]. Mutations in the UL27 region may also attribute to resistance to this drug [35]. 

### 3.5. Letermovir

Letermovir (LMV), the HCMV terminase inhibitor, has been recently approved for prophylaxis in stem cell HCMV-seropositive adult hematopoietic cell transplant recipients [36,37]. Resistance mutations are located mainly in UL56, and rarely in UL89 and UL51-terminase subunits, so there is no cross-resistance with other anti-HCMV drugs [38,39]. Clinical experience with LMV as a treatment for active HCMV infection is still limited, but the absence of myelosuppression, oral bioavailability and a good safety profile make LMV an eligible candidate for the treatment of HCMV infections that are resistant to approved agents or as an alternative to poorly tolerated intravenous options [40,41]. The use of novel molecular targets for the inhibition of HCMV replication, such as terminase complex, and exploring new options, like the viral alkaline nuclease, coded by the UL98 gene is the next step toward successful inhibition of HCMV replication. The creation of new therapeutics in order to develop more effective combination therapies would reduce the chance of the emergence of drug resistance [42,43,44]. 

## 4. HCMV Drug Resistance

The outcome of HCMV infection in an immunocompromised host, as well as congenital infection, significantly depends on the availability of antiviral therapy. However, there are considerable limitations of the currently registered drugs, such as poor oral bioavailability, associated toxicities and the potential for the development of resistance mutations. Quasispecies carrying resistance mutations to all currently available antiviral therapy have already emerged, partly because of the use of monotherapy in combination with low genetic barrier to resistance and due to the chronic persistence of HCMV infection in immunocompromised patients [45,46]. This emphasizes the need for alternative therapeutic targets, strategies and, of course, the development of an effective vaccine. One possible pathway is studying the distinct stages of the viral replication cycle in order to identify possible new drug targets. Combination therapy would most certainly reduce the probability of the emergence of resistance mutations, all while lowering the required dose, which would increase the tolerability. An alternative approach is to target a host cell protein or pathway that is essential for the completion of viral replication. While there is reasonable concern about the possible toxicity of this mechanism of treatment, it would reduce the possibility of the development of drug resistance [47,48]. 

## 5. HCMV Drug Resistance Testing

The emergence of drug-resistant forms is a great obstacle to the successful treatment of HCVM infection. Even with a selection of several antiviral drugs, clinical management of the infection is challenging due to its high frequency of drug resistance-associated mutations. The genes encoding the drug targets, UL54 (DNA polymerase) to GCV, CDV and FOS, UL56 (terminase complex) to LMV, and UL97 (phosphotransferase) to GCV and MBV, are usually the ones carrying resistance mutations [46]. However, for LMV, mutations in the genes that form the terminase complex, like UL89, can have an effect on its susceptibility to the drug [49]. Resistance gene mapping results showed that a single mutation in the UL27 gene is necessary and sufficient for resistance to MBV [35,50]. Drug resistance mutation maps for HCMV are regularly updated with recent information about newly detected polymorphisms that might cause resistance. They provide more detail on cross-resistance properties, and also emphasize the need to expand the regions covered in diagnostic testing. Therapy options for the treatment of HCMV infection are limited, so cross-resistance is a serious obstacle; however, there are few reports about multidrug resistance with mutations in both genes [51,52,53]. Patients should be tested for antiviral resistance whenever resistance is suspected, even after mutations have been identified, as additional resistance mutations can develop. It emphasizes the importance of proving that observed genetic changes confer resistance so that they can be distinguished from polymorphisms [46,54]. Normally, mutations in the UL97 gene occur initially, followed by UL54 mutation after a therapy switch. The appearance of a UL54 mutation alone without any detection of a UL97 mutation is rare. Interestingly, in a number of patients, the UL97 mutation could be detected exclusively in specific compartments, and not in blood. The manifestation of multidrug resistance is mostly associated with combined UL97/UL54 mutations [52,53].

Drug resistance to HCMV can be detected either by genotyping or phenotypically, using a virus grown in cell culture and applying various drug concentrations to it. Cell-associated plaque reduction assay detects drug resistance without the need for genetic information, but it takes time, requires a highly educated staff, is technically demanding and produces results that may vary significantly between different laboratories. Genotypic analysis can be performed without the need for viral isolation, which facilitates and speeds up the process; however, determining the degree of resistance when multiple mutations are detected may be difficult to deduce by genotypic testing alone. Therefore, the phenotypic studies are essential for analysis of the new mutations encountered in clinical isolates [46,54,55]. 

### 5.1. Sanger Sequencing

Sanger sequencing-based genotypic analysis is currently the most frequently used method by commercial reference laboratories. Restriction fragment length polymorphism (RFLP) and real time PCR assays can also be used for the detection of drug-resistant mutations. The limitations of the Sanger sequencing method are the strictly prescribed sample requirements, such as viral loads of at least 1000 IU/mL, for successful characterization and the substantial length of the fragment needed for analysis to cover all possible resistance mutation sites is also a challenge. It is necessary to read more than 2000 base pairs for the UL54 gene and about 1000 for the UL97 gene to detect resistance to GCV, VGV, CDV and FOS [46,51,52]. Fragments of 2100 base pairs of the UL56 gene and 800 base pairs of the UL89 gene need to be sequenced to determine resistance to LMV [48]. Resistance mutations to MBV are mostly covered with a nucleotide sequence of the UL97 gene; however, for complete resistance testing, sequencing of the UL27 gene is also required [35,50]. Only variants present in more than 20–30% of the overall viral population can be detected with Sanger sequencing, so the identification of low levels of resistance in a predominantly susceptible population, as well as mixed infections, may fail. Once a nucleotide sequence is read, it is compared to the wild-type referral strain in order to detect any polymorphisms. Only mutations that have a confirmed effect on the susceptibility of the antiviral drugs are reliable for the interpretation of the polymorphisms detected by genotypic methods [46,56]. Genotypic testing based on Sanger sequencing can be conducted in reference laboratories or as an in-house test, which some publications are demonstrating. Hall Sedlak et al. describe a rapid, sequencing-based assay for the UL97 and UL54 genes. This assay is performed in 96-well format with a single master mix and provides clinical results within 2 days. It sequences codons 440 to 645 in the UL97 gene and codons 255 to 1028 in the UL54 gene with a limit of detection of 240 IU/mL [57]. An in-house method for sequencing UL56 gene for detection of resistance mutations to LMV was described in a case report by Bosworth et al. [58]. The interpretation of detected mutations is also challenging but there are free, regularly updated internet algorithms that can be used for the detection of drug resistance mutations obtained by Sanger sequencing, such as Mutation Resistance Analyzer created by University of Ulm (https://dna.informatik.uni-ulm.de/software/mra/app/index.php?plugin=form, assessed on 8 January 2024).

### 5.2. Next-Generation Sequencing

Next-generation sequencing (NGS) technology offers a more sensitive, higher resolution view of emerging antiviral resistance capable of detecting minority variants down to as little as 1%, and it may be performed in samples with a lower viral load. However, cost per sample is still substantially high; NGS technology is not available at all laboratories; specialized skills are required for analysis; there is a scarcity of databases which summarize the clinically relevant antiviral resistance mutations for use in a bioinformatics pipeline; and detection of unexpected organisms or commensals of uncertain significance NGS assays is not widely used for the detection of HCMV antiviral resistance. Standardization of the method and diagnostic utility in comparison with traditional Sanger sequencing remains to be completed, but NGS is a powerful tool with a growing role in managing immunocompromised patients with suspected infection and is recommended for use in clinical trials. NGS can also provide data on the HCMV genotypes circulating in the population, which may facilitate the development of vaccines and immunobiological preparations, enable dynamic monitoring of risk groups (pregnant, newborns, children of the first year of life and patients who underwent solid organ transplantation), predict the epidemiological situation for cytomegalovirus infection, and improve the system of epidemiological surveillance of infections in general [59,60]. 

#### Evaluation of NGS as a Diagnostic Tool

There are a few evaluations of NGS as a diagnostic tool for detection of resistance mutations. In 2010, Schindele et al. demonstrated, using pyrosequencing, that the detection of GCV resistance-associated mutations occurring in the HCMV open reading frame of UL97 can be both fast and sensitive, with a minimum level of 6% mutant sequence variants. When compared to conventional dideoxy chain terminator sequencing, the method was more sensitive in detecting minor HCMV-mutant fractions in a wild-type population and, therefore, a useful tool for the early detection of emerging resistant mutations [61]. Nanopore sequencing was used by Li et al. to determine the complete genomes of HCMV in high-viral-load clinical samples without viral DNA enrichment, PCR amplification or prior knowledge of the sequences. Their data prove that improvements have been made and that, compared to Illumina, the final genomes from a urine sample and a lung sample achieved 99.97 and 99.93% identity and that Nanopore sequencing is capable of determining HCMV genomes directly from high-viral-load clinical samples with a high accuracy [62]. Garrigue et al. demonstrated that NGS technology allows a deeper discrimination of the emergence and persistence of a drug resistance mutation, which could be pertinent to the investigation of when routine Sanger sequencing detects only wild-type strains. Moreover, NGS-improved sensitivity helps in studying viral abundance, dynamics and diversity, which are less approachable with Sanger sequencing [63]. A dual-step NGS-based clinical assay that utilizes full-length gene amplification with a long-range PCR followed by shotgun sequencing for mutation analysis was developed by von Bredow et al. Their test achieved satisfactory performance with 96.4% accuracy, 100% precision and an analytical sensitivity of 300 IU/mL with a 20% allele frequency, showing that the implementation of a robust NGS LDT offers greater testing flexibility and sensitivity, accommodating a more diverse patient population [64]. Streck et al. compared NGS to Sanger sequencing and demonstrated two-test agreement for determining antiviral resistance/susceptibility and 88% (22/25) agreement at the level of resistance-associated mutations. The limit of detection of the NGS method was determined to be 500 IU/mL, and the lower threshold for detecting mutations associated with resistance was established at 15% [65]. The feasibility of the ViroKey^®^ SQ FLEX Genotyping Assay was assessed by examining 38 pediatric and 88 adult patient samples. The test proved to be most effective in detecting mutations in samples with a viral load above 1000 IU/mL, and it detected the 10 most important drug-resistant mutations, the most frequent being A594V, found in 5% of all tested samples [66].

## 6. Prevalence of HCMV Drug Resistance Worldwide

There are many published studies or case reports covering resistance to HCMV. Most publications are focused on resistance to GCV, since it is still the first-line therapy. Some are focused on a certain population, such as solid organ transplant (SOT) patients or hematopoietic stem cell transplantation (HSCT) recipients, and others are cover a more general population of viremic HCMV patients, but most studies include patients in which resistance to antiviral drugs is already suspected, which probably makes the percentage of resistance mutations detected somewhat higher. 

### 6.1. General Population of HCMV Viremic Patients

One of the most thorough studies assessing the prevalence of HCMV drug-resistant mutations among samples submitted for genotypic testing included 2750 patients analyzed in Eurofins Viracor’s reference laboratory for HCMV genotypic resistance testing from 1 January 2021 to 31 January 2023. Sequencing testing covered all currently available HCMV drugs and found that 826 (30.04%) of the samples tested had resistance to one or more of them. Resistance mutations were most frequently detected in UL97, with 27.64% of the samples carrying GCV mutations and 9.96% of samples carrying MBV mutations. The UL54 gene had 6.11% samples with drug resistance mutations to GCV, 5.98% to CDV and 1.76% to FOS. The UL56 gene had 7.17% of samples with mutations resistant to LMV detected, with mutations at codon 325 representing 80.95% of all observed mutations. Multiple drug resistance was detected in 250 samples; to 2 drugs in 215 samples and to 3 or more in 35 samples. The high prevalence identified in this study is proof that testing for drug resistance mutations only in samples already suspected to carry drug-resistant strains must be taken in context [67]. One of the studies that covers the general population of individuals with HCMV is Taiwanese research that chose 112 from 729 patients with HCMV syndrome for resistance testing. UL97 antiviral resistance mutations (L595S, M460I, and M460V) were found in four patients (3.6%) [68]. Another study by Shao et al. analyzed 40 clinical isolates in order to detect the drug-resistant mutations in the UL97 and UL54 genes. The results show that sequences obtained from the samples contained only polymorphisms (N685S, A688V, A885T, N898D in UL54; and D605E in UL97) and no mutations related to resistance to GCV [69]. 

### 6.2. Transplant Patients

HCMV infection is quite frequent in post-transplant recovery. The emergence of resistance mutations can additionally complicate treatment and lower the survival rate; therefore, papers on HCMV drug resistance in transplant patients are in abundance. An Australian study by Iwasenko et al. tested clinical specimens of transplant recipients (bone marrow, stem cell, kidney, heart, lung and liver) receiving antiviral prophylaxis, preemptive therapy or treatment for UL54 and UL97 gene antiviral resistance mutations. A greater diversity and number of UL97 and UL54 mutations were observed in heart and lung transplant recipients; whereas, antiviral-resistant HCMV infections in other transplant recipients were predominantly the result of a single mutant genotype. Mutations in the UL54 region were also more frequent in heart and lung transplant recipients (43%) compared to the 6% of HCMV-positive recipients of other transplanted organs or stem cells. HCMV strains containing previously unrecognized UL54 mutations (F412S and D485N) were also detected in one lung transplant recipient and one heart transplant recipient [70]. A Spanish nationwide study retrospectively analyzed 108 plasma samples from 96 HCMV infected transplant recipients. Drug resistance mutations were detected in 26.9% in the UL97 gene and 10.6% in UL54 gene. Mutations were either found exclusively in one of the genes, or patients carried mutations in both regions. L516R in UL54 and L397R/I and H411L in UL97 have been found for the first time in a clinical sample, and L595S/W was the most prevalent mutation [71]. A retrospective review of all patients with samples submitted for HCMV drug resistance testing at the national reference laboratory from 2011 to 2019 in British Columbia, Canada revealed that 27 out of 56 samples (48%) had resistance mutations detected. Most common were A594 (20%), H596 (12%) and L595 (12%) in the UL97 gene, which occurred more frequently in SOT patients than in HCST patients (58% vs. 27%, *p* = 0.05) [72]. 

### 6.3. HSCT

HCMV infection is a cause of serious complications in HSCT recipients. A retrospective study by Campos et al. included 22 subjects from a cohort of patients with different hematological malignancies submitted to allo-HSCT between 2010 and 2014 in Portugal. Resistance mutations were identified in seven patients. Five had resistance mutations in the UL97 gene, A594V, C592G, L595W and C603W, and two with resistance mutations in UL54, P522S and L957F. No simultaneous mutations were found and, while most mutations detected in this study are found quite frequently, mutation L957F was reported for the first time in a clinical specimen [73]. Chae et al. detected resistance variants in 10 (8.1%) patients, and variants of uncertain significance were found in 48 (39.0%) out of 123 patients with refractory HCMV viremia. Patients with any variant had a higher risk of severe graft-versus-host disease and lower one-year survival rates than those without (*p* = 0.003 and *p* = 0.044, respectively), and the presence of any variant reduced the rate of HCMV clearance. The data presented in this study emphasize the importance of identifying genetic variants associated with HCMV drug resistance in HCT (hematopoietic cell transplantation) recipients for providing appropriate antiviral treatment and predicting patient outcomes [54].

### 6.4. SOT

A single-center study by Young et al. included 2148 abdominal SOT recipients, with 116 patients in the study group treated for HCMV infection. Fourteen (12.1%, 0.65% of all SOT patients) had drug-resistant mutations [74]. Lopez-Aladid et al. prospectively assessed the presence of resistance mutations in a nation-wide study between September 2013 and August 2015. Resistance mutations were detected by Sanger sequencing in 9 out of 39 participants with all carrying mutations in the UL97 gene, and 2 patients also had one mutation in the UL54 gene. Resistance mutations contributed to incomplete suppression of the viral load and were more frequent in lung transplant recipients (44% *p* = 0.0068) and patients receiving prophylaxis after ≥6 months (57% vs. 17%, *p* = 0.0180) [75]. Van ler Buter et al. retrospectively investigated the prevalence of resistance-associated mutations and the factors associated with antiviral resistance in SOT patients with repeated high HCMV loads (>30,000 IU/mL) during antiviral treatment. Multiple samples from 113 SOT patients were tested by Sanger sequencing of the UL54 and UL97 genes, which showed resistance-associated mutations in 25 patients (22%), and a further 20 (18%) patients were shown to have mutations that were not known to be associated with antiviral resistance. High-level resistance mutations were most frequently seen in the UL97 gene. Several factors were associated with the development of resistance-associated mutations, suggesting the role of host immunity in human leucocyte antigen (HLA) mismatch [76]. A French multicenter prospective cohort enrolled 346 patients at initial diagnosis of HCMV infection and monitored them for ≥2 years. Sequencing of UL54 and UL97 genes and antiviral phenotyping was performed if viral replication persisted for >21 days of appropriate antiviral treatment. Resistance was suspected in 37 (10.7%) patients; 18/37 (5.2% of the cohort) had virological resistance, mostly due to a single UL97 mutation, but four cases of multidrug resistance were due to UL54 mutations. A high level of resistance or cross-resistance, leading to therapeutic failure can be attributed to isolates harboring UL54 mutations alone (1/5 of all samples with resistance mutations) or combined with UL97 mutations [77]. Silva Junior et al. presented a systematic review of the epidemiology, management, and burden of HCMV post-SOT in selected countries outside of Europe and North America in a 10-year period. The authors selected 49 studies out of 2708 studies and reported that rates of HCMV infection within one year were 10.3–63.2% (9 studies) and 0–19.0% (17 studies), up to 4.4% patients were resistant to treatment (3 studies) and no studies reported on refractory HCMV, which proves that rates of HCMV infection/disease post-SOT are highly variable [78]. A French two-center study included 81 SOT recipients with refractory HCMV infections monitored during a 10-year period and detected 24 genotypic profiles that conferred resistance to GCV and 2 to GCV and CDV. HCMV drug resistance influenced survival, causing mortality to increase to 19.2% from 3.6% [79]. 

#### 6.4.1. Kidney Transplant

HCMV mutations associated with antiviral resistance have become a major problem related to high mortality in kidney transplant patients. The incidence and outcomes of ganciclovir-resistant HCMV viremia was analyzed in 1244 kidney recipients transplanted from 2004 to 2008 in a Norwegian center. Ganciclovir-resistant mutations were detected in 27 patients (2.2%), of which 26 occurred in 209 HCMV IgG-negative recipients receiving a HCMV-positive kidney (12.5%). All mutations emerged in the UL97 gene, and none in UL54 [80]. A retrospective observational cohort study from Brazil included a total of 81 patients who underwent kidney transplantation between 2016 and 2018 and their HCMV viral load was monitored by qPCR. All 66 HCMV-positive kidney transplant recipients were analyzed by Sanger sequencing and resistance mutations were observed in 15 samples (22.72%). Most mutations were detected in the UL97 gene, while mutations in the UL54 gene were usually detected in combination with UL97 mutations, and in only two cases were UL54 mutations detected without UL97 ones. The resistance mutations identified in in UL97 were M460V, L595S, H520Q, two co-mutations, D465R + Del524 and A594P + D413A, and a 3-codon deletion (del598-601) [81]. 

#### 6.4.2. Lung Transplant

A single-center retrospective study investigated HCMV drug resistance in a cohort of 735 patients who received a lung transplant between January 2012 and October 2017. UL54 and UL97 genes were sequenced to detect mutations for 27 patients with HCMV viremia. From the cohort, 11 strains were resistant, 8 were sensitive and 8 were inconclusive. No differences in immunosuppression, acute rejection or pre-transplant sensitization were seen between case-matched groups. Drug-resistant HCMV infection was rare, but patients who developed it have decreased overall survival. Peak HCMV viral load and duration of HCMV viremia were associated with the development of a HCMV-resistant infection [82].

### 6.5. HIV-Positive Individuals

HCMV infection in HIV (Human immunodeficiency virus)-positive individuals presents a major health threat that can lead to pneumonia, intestinal and nervous system diseases and HCMV retinitis (where HCMV is disseminated to the eye and established productive infection results in retinal necrosis), which occurs in 25 percent of AIDS (acquired immunodeficiency syndrome) patients. Cytomegalovirus end-organ-disease (HCMV EOD) is also a major cause of debilitating illness in people living with HIV, especially in developing countries. The co-infection of HIV and HCMV creates a complex environment; HCMV accelerates HIV by reactivation where constant viremia is generated, and HCMV retinitis is associated with increased mortality. Preemptive therapy is recommended for patients receiving hematopoietic cell transplants and solid organ transplants to prevent the occurrence of HCMV EOD, but not for HIV-positive individuals due to concerns about the utility and safety of them. Treatment for HCMV infection is administered only when co-infection occurs. However, data on HCMV drug resistance in HIV-positive individuals are difficult to find [83,84,85]. A review by Azimi et al. compared data for subjects who received organ transplants and people suffering from AIDS in the period from 1980 to 2019. Overall resistance to GCV/VGV was 14.1%; however, in patients suffering from AIDS and organ transplantation, it was 19.5% and 11.4%, respectively [86]. Another study indicated that HCMV is more genetically variable in HIV-positive individuals. Sanger sequencing of the UL27 gene from 20 HCMV-positive HIV (9/20) and congenitally infected (11/20) patients revealed that K90 was the most prevalent polymorphism in both HIV-positive and congenitally infected patients along with polymorphisms Q54, D123 and R107, present in more than one sample. However, HIV-positive samples carried significantly more polymorphisms (*p* = 0.038) [87]. 

### 6.6. Children

HCMV is one of the main causes of congenital viral infection, with about 1 in 200 pregnancies affected and a variety of symptoms in the unborn child ranging from asymptomatic to death in utero. However, data on drug resistance in this patient population are rare and usually limited to case reports [88]. Choi et al. described a symptomatic congenitally infected infant treated with GCV/VGV, and, after a 50-fold increase in viral load after 6 weeks of oral therapy, analyses of HCMV sequences from both blood and urine demonstrated populations of viruses with M460V and L595F mutations in the UL97 gene. As expected, the analysis of the viral DNA retrieved from the newborn’s dried blood spot, which was taken before the start of treatment, found wild-type UL97 sequences [89]. Optimizing antiviral dosing according to weight gain in infants, as well as the constant surveillance of viral load is vital to prevent the emergence of drug-resistant variants, as demonstrated by Morillo et al. in the case of a symptomatic female infant. The patient developed ganciclovir resistance after 4 months of treatment, with increasing viremia and petechiae, and the treatment was stopped at 5 months of age [90]. Infants with immune deficiencies and HCMV infection require special attention. Two case reports describe infants with severe combined immune deficiency. Wolf et al. detected an unusual multiplicity of mutations in the UL97 substrate-binding domain. Resistant strains appeared within 10 days to 3 weeks from the initiation of therapy, which is earlier that in other individuals with HCMV [91]. Multiple mutations that conferred phenotypic resistance to all currently licensed systemic HCMV antivirals were detected in both UL97 and UL54 in an infant with autosomal recessive severe combined immune deficiency [92]. These findings indicate the need for early and frequent genotypic monitoring, as well as the rapid modification of therapy in this patient population. Kapman et al. describe a case of a preterm baby, born after 28 weeks of gestation, with disseminated congenital HCMV infection. The infant died on the 113th day of life but resistance-associated mutations in the HCMV UL97 gene were detected postmortem by pyrosequencing. Four different variants carrying resistance-associated mutations were found, each representing 11–17% of the total viral population [93]. Campanini et al. bring a case of HCMV drug-resistance in a symptomatic, congenitally infected newborn with an extremely complex virus population composed of a mixture of wild-type and multiple mutant GCV/VGV resistant strains carrying a variety of known mutations in the UL97 gene. Genotypic analysis detected four known UL97 single-nucleotide mutations (A594T/V, M460V/I, C592G), a new amino acid substitution (C607S), a new deletion (597–600) and the combination of M460V + A594V and M460V + C592G [94]. In a retrospective review of immunocompromised hosts with HCMV viremia between 2007 and 2017, Kim et al. included 31 patients who were 0–21 years old: 18 HSCT patients; 5 patients with primary immunodeficiency; 4 patients with malignancies; 3 heart transplantation patients; and 1 patient who was HIV-positive. Antiviral resistance testing of the UL54 and UL97 genes was performed on isolates from seven patients: five with persistent viremia and two who had not yet started antiviral therapy. Resistance was identified in three isolates from patients with common variable immunodeficiency (CVID) and recurrent Hodgkin’s lymphoma who had undergone autologous HSCT. Two had resistance mutations in both genes, while one isolate had resistance mutation only in the UL97 gene [95]. The complexity of virus populations circulating in congenitally infected patients might be due to the immature neonatal immune system and suggests a need for more frequent resistance testing, preferably by NGS technology [96,97,98].

### 6.7. LMV

LMV (Letermovir) is a relatively new antiviral, approved as prophylaxis for HCMV-positive stem-cell recipients in 2017. The viral terminase complex is formed with proteins pUL56, pUL89 and pUL51, but both in vivo- and in vitro-characterized resistance mutations are mainly located in the UL56 gene. The 2022 study by Chou et al. performed genotypic testing of the UL56 gene by Sanger sequencing in 1165 diagnostic specimens, detecting one or more LMV resistance mutations in 134 specimens. Most frequently, mutations were detected in codon 325 (C325Y/F/W/R), in 108 isolates, codon 369 (R369 S/G/T/K) in 13 isolates and V236M in 11, while mutations V231L, N232Y, Q234R, L257F and V363I were detected in 1–3 isolates each. Mutations in codon 325 mutations increased the LMV 50% inhibitory concentrations (EC50) 3000-fold which raises a concern for the therapeutic use of LMV, especially since this particular mutation is detected frequently in clinical practice [99]. Perchetti et al. analyzed drug resistance to LMV in 226 HCMV seropositive HCT recipients who received LMV prophylaxis from 2018 to 2020, and a HCMV viral load above 200 IU/mL was detected in 15 patients. The UL56 gene was successfully genotyped for the detection of LMV-resistant mutations in 7 patients, and a single resistance mutation, C325Y, was identified in an umbilical cord blood recipient [100]. Recio et al. genotyped plasma samples from 96 patients with a HCMV infection and detected no polymorphisms in the UL56 gene [71]. Santos Bravo et al. analyzed the emergence of de novo mutations in the UL56 terminase in transplant recipients with confirmed resistance to HCMV DNA polymerase and who were naive to LMV. The only UL56 variant detected by standard and deep sequencing was R246C, which is sensitive to LMV and was detected in 2/80 study participants, including one hematopoietic and one heart transplant recipient, which demonstrates the low rate of natural polymorphisms [39]. Muller et al. described a pUL51 natural polymorphism by sequencing 54 LMV-naive strains and comparing them to UL51 HCMV genes from 16 LMV therapy non-responders (prophylaxis or curative) where four unknown substitutions with no effect on viral fitness (D12E, 17del, A95V and V113L) were highlighted. The only resistance mutation in the UL51 gene was A95V, with a 13.8-fold increase in the LMV resistance level [101]. Chou et al. confirmed the contribution of mutation in the UL51 gene and P91S in vitro to resistance, both individually and to increasing the resistance level of mutations in the UL56 gene, S229F or R369M. A combination of UL56 mutations, S229F, L254F and L257,I when the P91S mutation was detected and confers a 290-fold increase in the resistance level [102]. Both of these studies stress the need to include the genotyping of the UL51 gene in clinical resistance testing, along with the UL56 and UL89 genes.

### 6.8. MBV

MBV (Maribavir) is not a newly designed drug, but it was only approved for the treatment of adult and pediatric refractory/resistant post-transplant HCMV infection in 2021, so data on MBV drug resistance are limited to those derived from clinical trials, and some case reports, almost exclusively on mutations detected in the UL97 gene. Sharing the target gene with GCV, some of the resistance mutations overlap as well [30,31,33]. A study by Chou et al. investigated mutations in viral genes, UL97, UL54 and/or UL27, in separate phase 2 trials, including 120 patients that received MBV for HCMV infection failing conventional therapy and that 119 received MBV for asymptomatic infection. HCMV viral load dropped below 200 copies/mL in 172 study participants during a course of 6 weeks. The genotyping data were available for 23 patients with reoccurring HCMV infection; while taking MBV, 17 had known resistance mutations (T409M or H411Y) and an additional 5 had UL97 C480F alone. Among 25 patients that did not respond to therapy at all, 9 showed T409M or H411Y mutations and 4 others showed C480F alone. C480F, detected for the first time in this study, had high-grade resistance to MBV and low-grade GCV resistance [34]. A study that originally revealed that the mutation in the UL27 gene may adapt the virus for growth in cell cultures in the absence of UL97 activity was conducted by Chou et al. in 2004. The strains carrying mutations in the UL27 gene had a lower level of resistance to MBV than the UL97 mutant [50].

## 7. Immunotherapy

The search for new therapeutic alternatives, like using virus-specific T-cells as biological anti-infection tools, is essential for circumventing the limitations of antiviral treatments. The effectiveness and safety of virus-specific T-cells in preventing and curing HCMV infection has been proven by different clinical trials. Another advantage is the possibility of creating T-cells specific to different viral infections which can reconstruct antiviral immunity in the immunosuppressed allo-HSCT recipient [103,104,105,106]. A different approach is using antibodies to protect against HCMV, using both neutralizing and non-neutralizing monoclonal antibodies which would also be a beneficial strategy for treatment of the immunocompromised target population [106,107]. 

## 8. Vaccines

The need for an effective HCMV vaccine for immunocompromised people undergoing SOT or HSCT, as well for the prevention of congenital HCMV transmission is of the most important public health interests; however, although there are many ongoing trials, no HCMV vaccine is licensed so far [108]. Obtaining sterilizing immunity against HCMV is very challenging, as reinfections occur in previously exposed individuals [109]. Both humoral and T-cell-mediated immunities are activated during HCMV infection, so it is expected that an effective vaccine should produce an immune response in both branches of adaptive immunity [110]. The first attempts to produce a HCMV vaccine date back to the 1970s with the development of live-attenuated viral strains. The attenuated Towne strain showed promising results in the reduction of severe symptoms of HCMV disease and graft rejection in kidney transplant recipients, but it failed to protect against infection [111]. To avoid any possible risk of replication and latency, replication-defective HCMV vaccines like V160 were produced. The efficacy of the V160 vaccine was tested by a two-dose or three-dose regimen in HCMV-seronegative healthy young women, and, although it was immunogenic, the administration of three doses failed to reduce the risk of primary HCMV infection [112]. Other approaches for vaccine design are focused on HCMV proteins, which are common targets of neutralizing antibodies, and important for the viral entry of gB, gH/gL/gO trimer and the pentameric complex gH/gL/UL128/UL130/UL131A [113]. The gB/MF59 vaccine, which based on monomeric gB adjuvanted with an oil-in-water emulsion of squalene, showed promising results in Phase 1 clinical trials; it was well tolerated and immunogenic [108]. The gB/MF59 vaccine proceeded to the next phase of research and, in two separate phase 2 trials conducted on healthy HCMV-seronegative women during first year after giving birth and adolescent females, showed 50% and 45% efficacy, respectively [114]. Better results were obtained in SOT recipients after vaccination with a gB/MF59 vaccine. In viremic patients after transplantation, both the viremia and duration of ganciclovir treatment decreased [110]. Numerous other strategies to find an effective HCMV vaccine, including plasmid-based DNA vaccines, RNA-based vaccines and peptide vaccines, are in different phases of clinical trials. So far, the most encouraging pathway to successful results includes the expression of various antigens in a single vaccine to achieve the stimulation of different branches of the immune system. Therefore, mRNA vaccines and vector vaccines are good candidates for that mission [108].

## 9. Conclusions

Successful treatment of HCMV infection remains a challenge. Direct-acting antivirals target only a few of the HCMV proteins, making the barrier for resistance low. The detection of resistance mutations in clinical settings is conducted either by in-house Sanger sequencing tests, sent to reference laboratories for testing or, less frequently, using the NGS method, and mostly for the first-line therapy such as GCV/VGV, and, consequently, for CDV and FOS. Providing timely HCMV drug resistance test results is crucial in order to design optimal patient-management strategies and minimize unnecessary exposure to second-line antivirals, as is making resistance testing for new antivirals available. The prevalence of the HCMV drug resistance is most thoroughly investigated in the population of transplant patients, with percentage of the detected mutations ranging up to almost 50% depending on the drug tested. Data on drug resistance prevalence in populations of HIV-positive individuals and congenitally infected children are rare, and usually limited to case reports which highlights the need to investigate these patient groups further. The future of the successful treatment of HCMV infection are rapid and sensitive drug resistance tests, searching for new targets for antivirals, the introduction of immunotherapy and the development of an efficient vaccine. 

## Data Availability

The data presented in this study are openly available in the references.

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
