# Peer review of "Human Cytomegalovirus (HCMV) Genetic Diversity, Drug Resistance Testing and Prevalence of the Resistance Mutations: A Literature Review"

_tropicalmed, 2024, doi:10.3390/tropicalmed9020049_

Round 1

Reviewer 1 Report

Comments and Suggestions for Authors

This is a comprehensive review focusing on the problem of drug-resistant human cytomegalovirus (CMV). The review is timely and informative, and I did not find any significant problems with the text.

I would like to make the following suggestions to further improve the manuscript:

1. One or more figures (e.g. diagrams) summarising the main points would make the complex content much more accessible to the reader.

2. The text is quite dense in parts, and splitting the longer subsections further into paragraphs would make it easier to read.

3. Regarding the genetic diversity of CMV and the number of viral ORFs (subsection 2), PMIDs 37459519 and 23180859 should be considered.

4. I believe GCV and VGV are not acyclovir derivatives (line 69).

5. The text should be proofread for minor language errors.

Comments on the Quality of English Language

There are some minor language problems, such as missing/superfluous/inaccurate articles ("the", "a/an"), punctuation errors (e.g. line 128), and extra spaces (e.g. line 67), but overall the English is of a very good standard.

Author Response

Dear reviewer 1,

We would like to thank you for the suggestions for improvement of our manuscript. The changes you requested have been marked in blue.

  1. One or more figures (e.g. diagrams) summarizing the main points would make the complex content much more accessible to the reader.

Since we don’t have any original data, any diagram would be repeating what was already stated in the text. However, if the reviewer insists, we could perhaps make a table of the genes affected by the resistance mutations and drugs that might become ineffective if a mutation emerges in the certain sequence. Off course we will take any further suggestions given by the reviewer.

  1. The text is quite dense in parts, and splitting the longer subsections further into paragraphs would make it easier to read.

The text has been split in the following sections:

Line 68: 3.1. GCV and VGV

Line 80: 3.2. FOS

Line 88: 3.3. CDV

Line 93: 3.4. Maribavir

Line 101: 3.5. Letermovir

Line 216: 5.2.1. Evaluation of NGS as a diagnostic tool

Line 258: 6.1. General population of HCMV viremic patients

Line 280: 6.2. Transplant patients

Line 305: 6.3. HSCT

Line 322: 6.4. SOT

Line 360: 6.4.1. Kidney transplant

Line 376: 6.4.2. Lung transplant

Line 508: 7. Immunotherapy

Line 518: 8. Vaccines

Line 551: 9. Conclusion

  1. Regarding the genetic diversity of CMV and the number of viral ORFs (subsection 2), PMIDs 37459519 and 23180859 should be considered.

PMID 37459519 is already cited as reference number 8:

Charles OJ, Venturini C, Gantt S, Atkinson C, Griffiths P, Goldstein RA, Breuer J. Genomic and geographical structure of human cytomegalovirus. Proc Natl Acad Sci U S A. 2023 Jul 25;120(30):e2221797120.

PMID 23180859 has been included in the text as reference number 9.

Stern-Ginossar N, Weisburd B, Michalski A, Le VT, Hein MY, Huang SX, Ma M, Shen B, Qian SB, Hengel H, Mann M, Ingolia NT, Weissman JS. Decoding human cytomegalovirus. Science. 2012 Nov 23;338(6110):1088-93.

All other references have been shifted accordingly. The reference number 103 has been removed altogether since it was an accidental duplication of the reference number 104:

  1. I believe GCV and VGV are not acyclovir derivatives (line 69).

The statement in question has been removed.

  1. The text should be proofread for minor language errors.

There are some minor language problems, such as missing/superfluous/inaccurate articles ("the", "a/an"), punctuation errors (e.g. line 128), and extra spaces (e.g. line 67), but overall the English is of a very good standard.

We have corrected punctuation and extra space errors in the lines 67 and 128, for the rest, since you find the English language to be of a good standard, we believe it can be corrected during the preparation of the manuscript for the print, off course, if the Tropical medicine and infectious diseases decides to publish it.

Best regards

Ivana Grgić and Lana Gorenec

Reviewer 2 Report

Comments and Suggestions for Authors

Human cytomegalovirus infection can cause serious complications in immunocompromised patients, including transplant recipients. Drug-resistant HCMV infections in these patients are major concerns. Doctors must first accurately diagnose HCMV infection and then identify the most effective antiviral drug. This review of Ivana Grgic and Lana Gorenec “CMV genetic diversity, drug resistance testing and prevalence of the resistance mutations: a literature review” examines therapeutic approaches against HCMV infection, the prevalence of drug resistance of different HCMV strains worldwide, and the development of genotypic techniques to rapidly diagnose drug resistance mutations. These techniques are particularly important because only rapid diagnosis of HCMV mutations at the point of care will allow timely life-saving intervention. An early diagnosis would allow the development of more effective combined therapies in reducing the possibility of the emergence of HCMV drug resistance.

General comments:

The manuscript is well written and understandable to a specialist readership.

In general, the organization and structure of the article are satisfactory and in accordance with the journal's instructions for authors. The title clearly indicates the focus of the review and the Abstract section well summarizes the article contents. The introduction provides sufficient background, and the other sections include results clearly presented and exhaustively analysed. The subject is adequate with the overall scope of Tropical Medicine and Infectious Disease, Special Issue: Genetic Diversity of Viruses: From Source Tracing to Treatment Tailoring.

Minor comments:

In the title of the article the virus has the acronym CMV but in the text, except in the lines 246, 251, 309, 344, 353, 354, 369-371, 374, 377, 382, 507, 508, 510, 512, 514, 516, 517, 519, 520, 526, 530, it is referred to as human CMV and the acronym is HCMV. Please make the text uniform.

Lines 138, 139 

“… but there are only few reports about multidrug-resistance with mutations in both genes”.

Please add references [51,52 and 54] 

Line 179

Please change “Sedlak” with “Hall Sedlak” and add reference number [56]

Line 309

Please add (hematopoietic cell transplantation) after “HCT”.

Lines 322, 323

Please change “solid organ transplant (SOT) “with SOT (see line 244) 

Line 329

Please change “human leucocyte antigen ALA)” with human leukocyte antigen (HLA)

Line 330

Please delete “Hantz et al” and add reference number [76] after the sentence “A French multicenter prospective cohort enrolled 346 patients at 330 initial diagnosis of CMV infection and the monitoring was continued for ≥2 years.“ (line 331).

Line 350

Please delete “Myhre et al” and add reference number [79] after the sentence “The incidence and outcomes of ganciclovir-resistant HCMV viremia was analyzed in 1244 kidney recipients transplanted from 2004 through 2008 in a Norwegian center. “ (line 352).

Line 373

Please add (Human immunodeficiency virus) after “HIV”.

Line 443

Please add (Letermovir) after “LMV”.

Line 477

Please add (Maribavir) after “MBV”.

Line 497

Please change “….anti-infection tools is a is essential for circumventing…..” with “….anti-infection tools, is essential for circumventing…..”

Author Response

Dear reviewer 2,

we would like to thank you for the comments and the opportunity to improve the manuscript. The changes made according to your comments are marked by color red.

In the title of the article the virus has the acronym CMV but in the text, except in the lines 246, 251, 309, 344, 353, 354, 369-371, 374, 377, 382, 507, 508, 510, 512, 514, 516, 517, 519, 520, 526, 530, it is referred to as human CMV and the acronym is HCMV. Please make the text uniform.

We thought it would be more appropriate to change the acronym in the title. However, if you find it more suitable, we will change it throughout the text.

 Lines 138, 139

“… but there are only few reports about multidrug-resistance with mutations in both genes”.

Please add references [51,52 and 54]

The references you are referring to are :

Göhring K, Wolf D, Bethge W, Mikeler E, Faul C, Vogel W, Vöhringer MC, Jahn G, Hamprecht K. Dynamics of coexisting HCMV-UL97 and UL54 drug-resistance associated mutations in patients after haematopoietic cell transplantation. J Clin Vi-rol. 2013 May;57(1):43-9.

Göhring K, Hamprecht K, Jahn G. Antiviral Drug- and Multidrug Resistance in Cytomegalovirus Infected SCT Patients. Comput Struct Biotechnol J. 2015 Feb 10;13:153-9.

And

Fischer L, Imrich E, Sampaio KL, Hofmann J, Jahn G, Hamprecht K, Göhring K. Identification of resistance-associated HCMV UL97- and UL54-mutations and a UL97-polymporphism with impact on phenotypic drug-resistance. Antiviral Res. 2016 Jul;131:1-8.

So the order of the references has been changed and the references in question are now under numbers 51, 52 and 53 and have been marked in the text accordingly. All other changes in the references are also marked red in the text, since Reviewer 1 asked for an additional reference that has been placed under number 9.

Line 179

Please change “Sedlak” with “Hall Sedlak” and add reference number [56]

The requested change has been made, but the reference in question is now number 57, now in line 187.

Line 309

Please add (hematopoietic cell transplantation) after “HCT”.

The requested change has been made, now in line 320.

 Lines 322, 323

Please change “solid organ transplant (SOT) “with SOT (see line 244)

The requested change has been made, now in line 333.

 Line 329

Please change “human leucocyte antigen ALA)” with human leukocyte antigen (HLA)

The requested change has been made, it is now in line 340.

 Line 330

Please delete “Hantz et al” and add reference number [76] after the sentence “A French multicenter prospective cohort enrolled 346 patients at 330 initial diagnosis of CMV infection and the monitoring was continued for ≥2 years.“ (line 331).

The requested change has been made in line 340, but the reference in question is now under number 77, in line 349.

 Line 350

Please delete “Myhre et al” and add reference number [79] after the sentence “The incidence and outcomes of ganciclovir-resistant HCMV viremia was analyzed in 1244 kidney recipients transplanted from 2004 through 2008 in a Norwegian center. “ (line 352).

The requested change has been made, but the reference in question is now under number 80, in line 367.

Line 373

Please add (Human immunodeficiency virus) after “HIV”.

The requested change has been made, now in line 386.

 Line 443

Please add (Letermovir) after “LMV”.

The requested change has been made, however, the abbreviation is already in the text, line 102.

Line 477

Please add (Maribavir) after “MBV”.

The requested change has been made, however, the abbreviation is already in the text, line 94.

 Line 497

Please change “….anti-infection tools is a is essential for circumventing…..” with “….anti-infection tools, is essential for circumventing…..”

The requested change has been made, in line 510.

The reference

Jalili A, Hajifathali A, Mohammadian M, Sankanian G, Sayahinouri M, Dehghani Ghorbi M, Roshandel E, Aghdami N. Vi-rus-Specific T Cells: Promising Adoptive T Cell Therapy Against Infectious Diseases Following Hematopoietic Stem Cell Transplantation. Adv Pharm Bull. 2023 Jul;13(3):469-482.

Has accidentally been copied, so one was removed from the text

Best regards

Ivana Grgic and Lana Gorenec
